# Costing analysis of a digital first-line treatment platform for patients with knee and hip osteoarthritis in Sweden

B. Ekman[1], H. Nero[2,3]*, L. S. Lohmander[2,3], L. E. Dahlberg[2,3]

1 Department of Clinical Sciences, Malmö (IKVM), Lund University, Lund, Sweden, 2 Orthopedics, Department of Clinical Sciences, Lund (IKVL), Lund University, Lund, Sweden, 3 Arthro Therapeutics, Malmö, Sweden

* hakan.nero@med.lu.se

**Data Availability Statement:** All relevant data are within the manuscript and its Supporting Information files.

## Abstract

Osteoarthritis (OA) constitutes a major and increasing burden on patients, health care systems and the broader society. It is estimated that around a quarter of the adult population is affected by OA in the knee and hip and that the prevalence of OA will increase over the coming decades largely due to aging and adverse life-style factors. Prevention and effective care are critical to manage the challenges posed by OA. Digital technologies offer opportunities to deliver cost-effective care for chronic diseases, including for OA. We report the results of a costing analysis of a new digital platform for delivering first-line care including disease information and physiotherapy to patients with OA and compare this with an existing face-to-face model of treatment. Both models are in accordance with National Treatment Guidelines in Sweden. The results show that overall the digital model costs around 25% of the existing face-to-face model of care. Based on existing evidence on the effects of these models, our findings also suggest that the digital platform offers a cost-effective alternative to the existing model of OA care. Depending on the extent to which the digital model substitutes for the existing model of care, significant resources can be saved.

## Introduction

Osteoarthritis (OA), the most common joint disease, mainly affects the knees and hips. Estimates for Sweden suggest that every fourth person is afflicted by OA and that it will increase in prevalence due to aging, obesity, and other contributing life-style factors [1]. In terms of global burden of disease, OA is the 11th out of 291 diseases to cause disability [2] whereas it in the U.S. is the fifth leading cause of disability [3]. Estimates of the economic burden of musculoskeletal diseases suggest OA economic impact to be as high as two percent of the gross domestic product (GDP) in industrialized countries, of which the largest direct costs include those for medication and surgery. The indirect costs, i.e. lost income, reduced productivity and spending on home care can reach as much as 4 600 USD per person annually [3]. A recent cost-of-illness study of OA estimated the yearly per person cost of OA at around 10 000 EUR

**Funding:** This study was financed by research grant funds from Vinnova - Sweden's Innovation Agency (grant number: 2016-04187, www. vinnova.se) and Stiftelsen för Bistånd åt Rörelsehindrade i Skåne (grant number: 2019-01-20, www.stiftbistandskane.se) to LED at the Department of Clinical Sciences, Lund, Orthopaedics, Lund University, Sweden. Arthro Therapeutics® (AT), the owner of Joint Academy®, provided part of the data and information used in the analysis. The funder AT provided support in the form of salaries for authors [HN, LSL, LED], but did not have any additional role in the study design, data collection and analysis, decision to publish, or preparation of the manuscript. The specific roles of these authors are articulated in the 'author contributions' section.

**Competing interests:** I have read the journal's policy and the authors of this manuscript have the following competing interests: HN is part-time employed and LED is the co-founder and Chief Medical Officer of Arthro Therapeutics®, the owner of Joint Academy®. LSL is a part-time consultant at Arthro Therapeutics®. This does not alter our adherence to PLOS ONE policies on sharing data and materials. No other relationships or activities exist that could appear to have influenced the submitted work.

(10 800 USD) [4]. Understanding the relative cost and effectiveness of available treatments and preventive measures would therefore be of considerable policy relevance [5].

The development of digital technologies across the healthcare sector by means of digital platforms may provide the opportunity to deliver evidence-based first-line care in accordance to global guidelines to patients at lower costs compared with traditional models of care. Cost advantages are likely to exist for patients, for health care providers, as well as to the broader society [6–8]. Other advantages of digital telehealth innovations are user flexibility, engaging asynchronous support from health professionals and the ability to receive care at home thereby avoiding travel. Specifically, treating chronically ill patients may have the potential to be most cost beneficial for all stakeholders. Accordingly, some recently developed digital platforms include managing type II diabetes, hypertension and musculoskeletal disorders [9, 10]. One of these platforms was developed to manage patients with OA of the knee or hip [11–13].

The aims of the present study were to assess the cost of providing digital care and best practice face-to-face care, and to compare the two models to evaluate the differences in resource use. Evaluating the resources required to deliver the alternative models of OA care would provide information for policy making. The study adopted a societal perspective by assessing all resources needed to deliver an episode of care to the patients. In particular, the analysis included costs on the health system and on the patient side. The study also measured the costs associated with carbon dioxide ($CO_2$) emissions due to transportations undertaken by the patients. Based on the results of the costing analysis and on existing evidence on the effects of the two models of OA care, the incremental cost-effectiveness ratio (ICER) was also computed. We also performed an analysis of the expenditure implications of scaling-up the most cost-effective model of care.

## Methods

The costing analysis compares the resources needed to deliver care with a digital OA treatment program, the Joint Academy® platform (JA) (www.jointacademy.com) [13], with the best practice face-to-face treatment, the 'Better Management of Patients with Osteoarthritis program', or the BOA model [14]. Both models provide individually tailored first-line management programs including disease information and exercises for patients that have been diagnosed with knee or hip OA. Patients self-select to receive care in either the traditional model or the digital model of care. Data suggest that there are no significant differences in age or sex between the two models of care; mean age is around 65 years of age and around 75 percent are women in both models [13, 14]. In presenting the results of the study, the CHEERS recommendations for reporting the results of an economic evaluation have been followed [15].

The costing analysis adopted the general approach of identification, quantification, and valuation of the cost items for both models [23]. The perspective of the analysis is that of the society. Data were collected from the providers of the care and did not include identifiable patient related data of any kind, hence did not require ethical approval.

### Better Management of Patients with Osteoarthritis (BOA)

The evidence-based face-to-face treatment model of a patient who has been diagnosed with OA of the knee or hip in Sweden, the BOA program, was initiated in 2008. According to Swedish National Guidelines a person diagnosed with knee or hip OA should receive a recommendation to register into one of the around 600 care units delivering the BOA-program [16]. In practice however, only around half of the Swedish hip or knee OA patients receive such a recommendation [17, 18].

The BOA model is consistent with international guidelines [19–21] and involves several standardized activities, including two to three one-hour, physiotherapist-led face-to-face lectures with information about the condition and available treatments. One additional session involves information by a former patient (1 hour; 44% participate in such a session). The patient is then offered individually tailored one-hour group exercise sessions twice weekly led by a physiotherapist over a period of 6–12 weeks. Around 60 percent of patients that continue in the program for at least 12 weeks participate in such sessions [17]. In all, a typical episode of treatment in the BOA-model involves 16 hours of provider (physiotherapist and a co-patient) contact time. The patient may also receive care from an occupational therapist in case of need. Patients are followed-up three and twelve months after completing an episode of care in terms of mobility, pain, and health-related quality of life (HRQoL using the EQ-5D-5L instrument).

The BOA model also involves additional resources for the clinic, the patient, and others, including planning and preparation of sessions, transportation to and from the site, direct costs (user-fees), and time off work for patients who are employed. In addition, physiotherapists who would like to qualify for the BOA program are required to take a one-day course led by a senior physiotherapist.

## Digital model

The digital model was inspired by BOA and is an alternative delivery of care, likewise based on evidence and global guidelines. It consists of a patient interface that provides individually tailored information on OA and exercises for rehabilitation, and support for life-style changes. It includes a provider interface where a trained physiotherapist can follow progress of the patient and provide feedback and support throughout the treatment period. Exercises are distributed daily, with instructional videos including graphical elements coupled with text instructions, to ensure proper execution.

The model has shown significant effects on key indicators including mobility, pain, and physical function in recent studies [11–13, 22]. The digital model is initiated by the patient providing key information about his or her condition into the system platform. The information is reviewed by a physiotherapist who then contacts the patient via the application to confirm the OA diagnosis. During that visit, the patient is able to ask for any additional information about the treatment model or his or her particular concerns related to the condition. Specifically the following treatment contacts are identified as constituting the regular set of physiotherapy activities and interactions during a 12 week period (duration in minutes; means of interaction): start-up meeting (15; telephone); daily coordination and adjustment (varies as needed; digital platform); weekly follow-up (5–8; digital platform); 6-week follow-up meeting (15; telephone); monthly follow-up session (5–8; digital platform); 3-month follow-up (15; telephone); additional interactions (as needed). In all, a typical episode of care consists of at least 18 activities that take around 143 minutes (2.38 hours) to perform over a period of care of 12 weeks (based on the Terms of References for physiotherapists by Joint Academy®).

In contrast to the BOA program, the digital model of care is open-ended and continues as long as the patient's condition improves, until the physiotherapist deems that behavior change (the participant is exercising regularly and will continue to do so without support) has been achieved or until other treatment is needed, such as surgery (total joint replacement).

Similar to the BOA model of care, the digital program requires other resources, including preparations and follow-up on the part of the provider and the patient. In addition, in order to provide care through the digital platform, the physiotherapist is required to take a mandatory online training course in the use of the platform and a short course in online physiotherapy

provision. Finally, the physiotherapist is required to pass an online certification exam. These training and exam events take a total of two hours.

## Analysis

The analysis estimates the resources needed to deliver an episode of OA care (the unit cost) by either model in 2018, the most recent full year for which data were available. Since both models provide individually tailored regimens there is significant variation across patients with respect to the scope and intensity of the treatment episodes. To ensure a fair comparison between the models, an episode was defined as care over a 12 week-period for both models. Furthermore, care is taken to avoid over- or underestimation of resource use by adopting a conservative approach to the quantification and valuation in cases when use can only be estimated by means of inexact methods, such as transportation time and technical support costs.

In the first step of the analysis each cost item was listed across three main domains to reflect the societal perspective of the analysis: the health care system (i.e. clinic or provider), the patient, and other sectors of society. Identification was done by reviewing documents that describe the two models and by consulting experienced users of the two models of care. Using the same sources of information, each cost item was quantified in terms of time or other resources needed to deliver the care. Finally, valuation was done by consulting relevant sources of information for the particular cost item, such as mean gross hourly wage rates (of physiotherapists and the general public). The table below lists the main cost items for each domain and describes how they have been quantified and valued (Table 1).

Time is valued according to the human capital method using average gross hourly wage rates for physiotherapists and the general population obtained from Statistics Sweden wage statistics [23]. Providers' time is valued including non-wage social fees set at the legally mandatory minimum rate of 31.42 percent of gross wage. Patients' time is valued at the reference value of leisure of 30 percent of the gross wage rate and net of any social fees. Analysis of when during the day patients are active in JA show that they predominantly log in during the morning or in the evening without any difference between the two groups in terms of age. No adjustment was made for when patients in the JA model perform their training exercises as compared with the BOA model. To account for the cost of facility rent a ten percent surcharge is added on the hourly value of staff time in the BOA model of care.

In both models of care, the patient undergoes a set of instructional lessons and exercise sessions. As described above, these are face-to-face sessions in the BOA model of care and online based in the digital model. An important difference between the models is that the sessions are group-based in the BOA model. This means that to obtain the unit cost of care, these costs are divided by the average number of participants. From a payer perspective, however, the costs for a physiotherapist and office-space remain the same regardless of the number of participants in a group session. Consequently, these costs are reported separately in order to obtain a comprehensive cost profile of the models. In addition, when adjusting the time-period for the BOA model, the introduction and information sessions are only counted once as these are independent of the length of treatment.

In addition to the time costs associated with the exercise sessions resources are also needed for preparatory and follow-up activities. In the BOA model they involve preparing the training facility, arranging equipment, and booking patients. In the JA program they mostly involve reading up on the patient's reporting data and preparing responses to any particular question or issue that the patient may have raised in his or her reports. These resources are reported separately as administration costs that have been measured by consulting physiotherapists

**Table 1. Identification, quantification, and valuation of main cost items.**

| Identification of cost item by domain | Quantification | Valuation | Source |
|---|---|---|---|
| **A) Provider/Clinic/Health system (BOA and JA)** | | | |
| **Contacts/Visits** | Estimate number and duration in hours of contacts/visits per episode of care | Mean gross hourly wage by professional (physiotherapist), including non-wage social fees (50.1%) | Statistics Sweden– www.scb.se |
| Introduction | | | JA and BOA[1] |
| Training/Rehabilitation session | | | JA and BOA |
| JA-contacts (asynchronous) | | | JA and BOA |
| **Administration** | | | |
| Technical support (JA) | Budgeted amount/number of patients | | JA |
| Preparations/Follow-up | Share of contact/visit duration | Mean gross hourly wage by professional (physiotherapist) | Statistics Sweden– www.scb.se |
| **Training education** | Training of physiotherapists in technical application and digital care | Mean gross hourly wage by professional (physiotherapist) | Statistics Sweden– www.scb.se |
| Training | | Mean gross hourly wage multiplied by 1.5 | JA and BOA |
| **B) Patient** | Estimate number and duration in hours of contacts/visits per episode of care | National mean gross hourly wage, net of non-wage social fees | Statistics Sweden– www.scb.se |
| Introduction | | | JA and BOA |
| Training/Rehabilitation session | | | JA and BOA |
| JA-contacts (asynchronous) | | | JA |
| **Administration** | | | |
| Preparations/Follow-up | Share of contact/visit duration | National mean gross hourly wage | JA and BOA |
| Transportation to and from clinic | Average distance to clinic | | Riksrevisionen rapport 2014:22; Bilaga 1 – Analys av närhet till vårdcentral |
| **Direct costs** | SEK | | |
| User-fees | | | |
| **C) Other** | | | |
| $CO_2$-emissions | Estimate length of transportation | Calculate $CO_2$-emissions | Emisso; http://www.utslappsratt.se/berakna-utslapp/berakning-av-utslapp-fran-bilar/ |

[1] Consultations with key informants of the respective care model. BOA—Better Management of Patients With Osteoarthritis; JA—Joint Academy[®]; $CO_2$ –Carbon dioxide; SEK—Swedish kronor.

from both models of care who were able to provide estimates of the time required for these supporting activities.

Physiotherapists in the JA program are mandated to undergo formal training and to pass a test in order to obtain the required certificate to receive patients in this program. The training program involves three separate sessions: a 20-minute self-learning session on general OA care, a 40-minute self-learning session on technical and care-related aspects of providing OA care over a digital platform, and a final one-hour JA-staff supported test involving vignette like situations of digital OA care. As these types of costs are one-off activities they are reported separately in the results section. As also noted above, the BOA program also requires participating physiotherapists to take a one-day training course. The costs of these training events are estimated and reported below.

The digital model of care requires a certain amount of technical support, both to physiotherapists and to patients. Such support is provided as needed on a stand-by basis. To quantify the unit cost of this item the total annual cost of support is divided by the total number of patients in 2018. While it is likely that also providers in the BOA model of care require a certain amount of technical and other types of support, no information and data on such support have been obtained and it is therefore assumed that the total cost of technical and other support in this model is equal to half of that of the digital model.

Transportation costs for the BOA group of patients are estimated by multiplying driving time based on average distance to a health care clinic in Sweden with the average number of appointments. This estimate is based on a recent analysis by the Swedish National Audit Office of the distance and travel time to a primary care clinic by the general population [24].

Vehicle transportations are assumed to lead to $CO_2$-emissions [25]. While the transportation mode varies, it can be assumed, given the debilitating nature of OA, that the majority of transportations is made using a motor vehicle (car or bus). Finally, it is assumed that all patients reach the national user-fee ceiling of 1,100 SEK per year in direct financial costs.

The analysis does not consider costs for research and development and any other investment costs. The main reason for this omission is that such costs are largely unknown for the BOA model of OA care, which has been in effect more than a decade and developed over a similarly long period of time. Finally, no costs for pharmaceuticals have been included as medicines are not part of the standard physiotherapy treatment regimen in either of the programs.

### Incremental cost-effectiveness ratio

Based on the results of the costing analysis and on the results of one previous study [12] on the effects of the digital care model, the incremental cost-effectiveness ratio (ICER) was also calculated. For the given cost and effect differences, the ICER shows the cost per effect unit of adopting the intervention compared with the existing treatment model [26].

## Results

Based on the estimates of the resource domains, the results of the analysis show that the most common resource is time used for various care activities, including training/rehabilitation sessions, preparations and follow-up, and transportation (Table 2). For a complete table of costs, see Supplementary S1 Table.

From a societal perspective, delivering one episode of care to a patient digitally costs 2 776 SEK compared with 10 610 SEK for a face-to-face patient, a difference of 7 835 SEK. In both models of OA care, the largest costs are borne by the patient, particularly so in the BOA model where 87 percent of total societal costs fall on the patient, compared with two-thirds in the digital model. While the largest cost item for the patient in the digital model is direct financial cost in the form of user fees, such costs constitute the smallest cost item in the BOA model. Conversely, due to the on-sight nature of care in the BOA model, the patients' largest costs include the time spent on performing the sessions and on transportation to and from the clinic. From the patient perspective, a critical difference between the two models is the ability to avoid transportation costs in the digital model of care.

Differences between the two models of care can also be viewed from the health care system perspective. The total unit cost of delivering an episode of care in the digital model is 766 SEK compared with 1 299 SEK in the BOA model, a difference of 534 SEK. As can be seen from Table 2, these costs are mostly driven by the training sessions, which are more frequent in the digital model but also considerably shorter. The administrative costs (preparations and follow-up) are higher in the BOA model compared with the digital model, even assuming that

**Table 2. Costs of standard treatment model (A: BOA) and digital model (B: JA).**

| Domain/ Item | Cost domain | A:BOA | | | | | B: Joint Academy | | | | |
|---|---|---|---|---|---|---|---|---|---|---|---|
| | | Amount /Number | Length, hrs | Total hrs | Value, SEK | Total unit cost, SEK | Amount /Number | Length, hrs | Total hrs | Value, SEK | Total unit cost, SEK |
| **A. System** | | | | | | | | | | | |
| | Contacts/Visits/ Sessions | 28 | | | | 619 | 18 | | | | 145 |
| | Administration | | | | | 651 | | | | | 610 |
| | Training of physiotherapists | | | | | 30 | | | | | 11 |
| | A: Sub-total | | | 56 | | 1 299 | | | 139,2 | | 766 |
| **B. Patient** | | | | | | | | | | | |
| | Contacts/Visits | | | | | 5 504 | | | | | 716 |
| | Administration | | | | | 1 204 | | | | | 195 |
| | Transportation | | | | | 1 445 | | | | | - |
| | Direct costs | | | | | 1 100 | | | | | 1 100 |
| | B: Sub-total | | | | | 9 253 | | | | | 2 010 |
| **C. Other** | | | | | | | | | | | |
| | C: Sub-total | | | | | 59 | | | | | - |
| | Total | | | | | 10 611 | | | | | 2 776 |

Source: Calculations based on study data. For a complete table, see S1 Table. SEK = Swedish krona, 100 SEK = 10.2 USD = 9.43 EUR.

technical and other types of support costs are only half of those in the digital model of osteoarthritis care.

Finally, the BOA program of care is estimated to lead to 0.014 tons of $CO_2$ emissions. The value of these is obtained using the current price of emissions rights from the European $CO_2$ emissions market, EU-ETS [27]. The total emissions amount to 133 tons based on an estimate that around 9 500 patients participated in a full episode of care in 2018. The price of one ton of $CO_2$ emission is around 220 USD resulting in a total cost of around 555 747 SEK in CO2 emissions due to transportation to and from the clinic in the BOA model of care.

## Cost-effectiveness analysis

In a recent study of the effect of the digital model of care, Nero and colleagues showed that patients with knee OA receiving care in the digital model report on average a reduction in experienced pain from 5.7 to 3.2 (a reduction by 2.5 points on a 0–10 scale, or a 44 percent reduction) after 12 weeks [13]. Patients with knee OA receiving the care in the BOA model report a reduction from 5.2 to 4.1 (a reduction by 1.1 points on a 0–10 scale, or a reduction of 21 percent) after the same amount of time [14].

Combining the results from the costing analysis with the results from the effect analysis an incremental cost effectiveness ratio (ICER) can be computed which shows the cost per unit of effect improvement [26]. The following ICER is calculated:

$$\text{ICER} = [\text{CostJA} - \text{CostBOA}]/[\text{EffectJA} - \text{EffectBOA}] = [2\ 776 - 10\ 611]/[3.2 - 4.1] = 8\ 705 \text{ SEK}$$

While there are no set thresholds to decide whether an ICER of this magnitude can be considered cost-effective [26], the combined findings suggest that the digital model of OA care is cost-effective compared with the standard model of care.

**Table 3. Expenditure savings if the digital model substitutes BOA.**

| Substitution rate | Total care cost if JA substitutes for BOA | Difference, SEK | Difference, % |
|---|---|---|---|
| 25% | 19 705 673 | 80 725 584 | 80 |
| 50% | 13 137 115 | 87 294 142 | 87 |
| 75% | 6 568 558 | 93 862 699 | 93 |

Source: Calculations based on costing results and BOA data. Assuming adherence rate of 60%.

### Total expenditure effects of the Joint Academy model

In 2018, 9 465 patients received care in the BOA model and 1 421 patients received care in the digital model for at least twelve weeks. The total societal costs of providing the BOA model of care was approximately 117 million SEK. The cost of the digital model was 4.1 million SEK. Table 3 presents the estimate of the expenditure savings that would occur if some share of the more costly BOA model of care is substituted for by the less costly digital model of care.

The estimates show that if half of all patients that received care by BOA in 2018 instead had received care in the digital model, around 87 million SEK would have been saved in direct total societal costs of OA care. These estimates are net of any resources saved or value gained by the estimated outcome differences between the two models, as well as other differences with respect to treatment complications, unnecessary diagnostics and surgeries and medications that may have occurred.

### Discussion

We have here shown that first-line OA treatment delivered digitally may cost as little as one-quarter of the traditional in-person care, with cost advantages both on the health system side and on the patient side, as well as on the side of the broader society. Most of the cost differences are found on the patient side as the face-to-face model imposes significant costs to the patients in terms of time and travel costs.

Understanding cost differences between alternative models of care is important for effective policy making. More generally, however, managing a common disease with increasing prevalence and significant economic burden on society and healthcare is a large and complex task. First-line management globally recommended in clinical guidelines for knee or hip OA includes disease information and exercise treatment [19–21]. Observational studies have shown this management to improve patient pain and function [28]. Widely used structured programs, such as Joint Academy®, BOA, and GLA:D® have in reports of observational data confirmed these beneficial results in real world settings. In addition to significantly improving patient pain and function and decreasing use of medications and sick leave, Joint Academy® and BOA also decrease willingness for surgery [13, 14, 29].

So far, no trials have been published comparing outcomes from OA face-to-face programs with digital equivalents. While care model preferences may often determine patient choice, other factors such as economics, flexibility, accessibility and scalability may be important as well, in particular to the health-care provider. Aside from cost, as shown here, digital programs differ from in-person care in several aspects. One of them is scalability and the economies of scale associated with digital models of care. User flexibility, instant on-demand access, engaging asynchronous support from health care professionals, and the ability to receive care at home thereby avoiding travel are other relevant aspects [30]. Equality in access to care between regions with or without easy access to in-person care may support more widespread implementation of first-line care for OA, as well decrease the need for transportation in connection

with care episodes. A digital program can with relative ease be translated and be implemented with similar quality in areas with different cultures and languages.

Economic gains by increasing the use of first-line OA management delivered digitally are not limited to lower costs of the first-line treatment for patient and provider. Routine OA care includes costly interventions, some of them shown to be of high patient value, others of doubtful or low patient value [31]. Total joint replacement is of high value for those with severe knee or hip OA, but not for all [32–34]. In the US alone, half a million hip replacements, and 1.1 million knee replacements are projected to be performed in 2020, at an estimated cost of between 30 and 48 billion USD [35]. First-line OA management has been shown to decrease patients' interest (willingness) in joint surgery [12, 14, 36]. If even a small proportion of those procedures was avoided or delayed through appropriate delivery of first-line OA management at the population level, cost savings would be considerable. Arthroscopic surgery of the knee or hip is one of the most common orthopedic procedures, but of contested patient value for those with degenerative joint changes [37]. As for joint replacement surgery for OA, preceding a shared decision on arthroscopic surgery with a structured first-line OA management may help avoid a considerable number of surgical procedures with ensuing annual cost savings of billions of USD [38–41].

Our study has some limitations. The cost-effectiveness comparison of care models was based on retrospective data while use of prospective cohort data or a randomized trial would be at lower risk of bias. Our study used outcomes and costs from Sweden and generalizability to other countries and health care systems will need to be confirmed. However, the face-to-face program GLA:D has shown similar outcomes for Denmark, Canada and Australia [42], suggesting that patient-relevant outcomes may be generalizable across countries. We obtained no data on the actual costs of any technical and other types of support for the BOA model of care. These were assumed to be half of those in the digital model. Removing completely these types of costs from the BOA model of care does not change the overall results in any material way. One of the largest cost items to the patient in the BOA model is transportation costs to and from the clinic. The quantity of these costs can only be estimated with some level of uncertainty, current projection is most likely an underestimate of travel time as the number of BOA clinics is less than half of the number of primary care clinics in the country [43]. Removing them altogether would most likely result in an underestimate of the patient costs as the face-to-face nature of that model of care does require the patient to spend some time and other resources getting to and from the clinic to perform the training and introductory sessions. The $CO_2$ emissions are estimated with some uncertainty. However, removing these would not change the overall findings in any material way given their relatively small impact on the total societal cost of osteoarthritis care by means of the BOA model. Finally, as noted above, patients self-select into either of the treatment models. This may lead to a risk of selection bias of the cost estimates. However, we also note that the two groups of patients are comparable in terms of sex and age, suggesting that this source of potential bias may be limited.

## Conclusion

This cost comparison and cost-effectiveness analysis of digital and face-to-face modes of delivery of structured first-line treatment for OA of the knee and hip suggests that digitally delivered care can substantially decrease the economic burden of OA for patients and health care providers. Digital OA care is a cost-effective alternative to existing on-sight models of care. Substituting existing care for digital care may lead to considerable savings for both patients and health systems. However, actual savings will be influenced by differences in OA management, reimbursement mechanisms and healthcare system characteristics across countries.

## Supporting information

**S1 Table. Table of costs, complete version.**
(XLSX)

## Acknowledgments

The authors would like to thank Ulf Tellgren at Abels Rehab/Region Skåne and Kristin Wetterling at BOA for information on the BOA program, and Oscar Pendse and Emelie Blennow at Arthro Therapeutics for helping with data from Joint Academy.

## Author Contributions

**Conceptualization:** B. Ekman, H. Nero, L. S. Lohmander, L. E. Dahlberg.

**Data curation:** B. Ekman.

**Formal analysis:** B. Ekman.

**Funding acquisition:** H. Nero, L. E. Dahlberg.

**Investigation:** B. Ekman, H. Nero, L. S. Lohmander, L. E. Dahlberg.

**Methodology:** B. Ekman.

**Resources:** L. E. Dahlberg.

**Visualization:** B. Ekman.

**Writing – original draft:** B. Ekman.

**Writing – review & editing:** B. Ekman, H. Nero, L. S. Lohmander, L. E. Dahlberg.

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
