## [Decision Letter · Decision Letter 0]

23 Jun 2020

PONE-D-20-13627

Costing analysis of a digital first-line treatment platform for patients with knee and hip osteoarthritis in Sweden

PLOS ONE

Dear Dr. Nero,

Thank you for submitting your manuscript to PLOS ONE. After careful consideration, we feel that it has merit but does not fully meet PLOS ONE’s publication criteria as it currently stands. Therefore, we invite you to submit a revised version of the manuscript that addresses the points raised during the review process.

We look forward to receiving your revised manuscript.

Kind regards,

Patrick Bergman

Academic Editor

PLOS ONE

Journal Requirements:

1. Please ensure that your manuscript meets PLOS ONE's style requirements, including those for file naming. The PLOS ONE style templates can be found athttps://journals.plos.org/plosone/s/file?id=wjVg/PLOSOne_formatting_sample_main_body.pdf and https://journals.plos.org/plosone/s/file?id=ba62/PLOSOne_formatting_sample_title_authors_affiliations.pdf

'I have read the journal's policy and the authors of this manuscript have the following competing interests: HN is part-time employed and LED is the co-founder and Chief Medical Officer of Arthro Therapeutics®. LSL is a part-time consultant at Arthro Therapeutics®. No other relationships or activities exist that could appear to have influenced the submitted work.'

We note that one or more of the authors are employed by a commercial company: Arthro Therapeutics®.

3. We note you have included a table to which you do not refer in the text of your manuscript. Please ensure that you refer to Table 3 in your text; if accepted, production will need this reference to link the reader to the Table.

Additional Editor Comments (if provided):

Reviewers' comments:

Reviewer's Responses to Questions

**Comments to the Author**

1. Is the manuscript technically sound, and do the data support the conclusions?

Reviewer #1: Partly

Reviewer #2: Yes

2. Has the statistical analysis been performed appropriately and rigorously? 

Reviewer #1: N/A

Reviewer #2: Yes

3. Have the authors made all data underlying the findings in their manuscript fully available?

Reviewer #1: Yes

Reviewer #2: Yes

4. Is the manuscript presented in an intelligible fashion and written in standard English?

Reviewer #1: Yes

Reviewer #2: Yes

5. Review Comments to the Author

Reviewer #1: Remote treatment has revolutionized many diseases as described in this retrospective study for osteoarthritis.

The authors should comment on the following:

The description of the technical setup at home is brief referring to previous reports only, e.g. are the instructions online or recorded videos? Do the patient show her movements for the physiotherapist who might provide feedback?

Since the cost estimations are retrospectively based, the selection and preference for different patient groups are not described and the subsequent impact on the results. This might overestimate the potential savings, e.g. the role of cognitive status of the patient might affect the compliance and requiring a physical visit.

Since the patient transportation and time are regarded as the largest costs, the authors should comment on that, e.g. retired patients have no cost for time, and on the choice of vehicle, e.g. electric car. The latter implies lower or no CO2-effect. Have you estimated the cost due to real emission or the compensation cost, which is quite different.

Regarding CO2-emissions, are the servers for the cloud based Joint Academy fossil free?

Would the patient make a concomitant errand when visiting the physiotherapist?

Discussion: You write "So far, no trials have been published comparing outcomes from face-to-face programs with digital equivalents." Do you refer the joint problems or other areas? In fact, there are numerous RCT-studies in depression - mostly from Sweden - treated with cognitive based therapy (CBT) on internet.

What is the impact of the platform vs the physiotherapist, which might influence the costs? Will self-care suffice for some patients, e.g. even cheaper?

To optimize the cost savings, how can patients be selected in the future?

In the declaration for conflict of interest, it should be mentioned that Arthro Therapeutics owns the JointAcademy Platform.

Reviewer #2: Manuscript Title: Costing analysis of a digital first-line treatment platform for patients with knee and hip osteoarthritis in Sweden

Overview: In this study the authors present the results of a costing tool that provides a comprehensive analysis of a new digital platform for delivering first-line care including disease information and physiotherapy to patients with osteoarthritis (OA) and compare this with an existing face-to-face model of treatment. This is an original study and it provides new and valuable information in the area of digital health. The manuscript is very well written and presented. The cost comparison and cost-effectiveness analysis of digital and face-to-face modes of delivery of structured first-line treatment for knee and hip OA strongly supports the idea that digitally delivered care can substantially decrease the economic burden of OA for patients and health care providers. Patients with OA can still benefit from face-to-face interactions with their health care providers but in terms of monitoring and follow-up digital tools are the way forward for the future.

This reviewer does not have any specific comments for major or minor revisions. The major papers that have been published on the global burden of OA have all been cited in the office have done a very good job of leading in this niche but very important area of cost effectiveness research.

6. PLOS authors have the option to publish the peer review history of their article (what does this mean?). If published, this will include your full peer review and any attached files.

Reviewer #1: No

Reviewer #2: Yes: Ali Mobasheri, University of Oulu, President of OARSI

---

## [Author Response · Author response to Decision Letter 0]

1 Jul 2020

Rebuttal letter

We thank both reviewers and the editor for taking the time to read and comment on our manuscript, so we may clarify certain aspects and improve its quality. Below you will find all reviewer comments followed by our response and if needed, our actions within the manuscript.

Reviewer #1

Reviewer comment: 

Remote treatment has revolutionized many diseases as described in this retrospective study for osteoarthritis.

The authors should comment on the following:

The description of the technical setup at home is brief referring to previous reports only, e.g. are the instructions online or recorded videos? Do the patient show her movements for the physiotherapist who might provide feedback?

Author response:

In terms of the exercises, they are distributed daily and made up of short text instructions, followed by a video with motion graphics and text instructions incorporated, where the exercise is shown, explained and details of relevance are highlighted. The exercises are prepared with great detail and can be replayed as many times as needed, to ensure that the patient has understood how to perform it properly. Currently, patients can not share video content with their physiotherapist using the app. However, when needed it is possible to chat with the physiotherapist if there are uncertainties or questions regarding the exercises or how they should be performed.

Author action:

Added the following text to Methods, page 6, paragraph 6:

“[…]Exercises are distributed daily, with instructional videos including graphical elements coupled with text instructions, to ensure proper execution.[…]”

Reviewer comment:

Since the cost estimations are retrospectively based, the selection and preference for different patient groups are not described and the subsequent impact on the results. This might overestimate the potential savings, e.g. the role of cognitive status of the patient might affect the compliance and requiring a physical visit.

Author response:

Thanks for highlighting this. Patients self-select into either of the two models of care. Hence as you state, it would be fair to mention the issue of self-selection as a possible source of bias. However we also note that there are no significant differences between the two groups of patients in terms of age and sex, which may indicate that this source of potential bias may be limited.

Author action:

Added text in the Method, first paragraph, page 4: “Both models provide […]. Patients self-select to receive care in either the traditional model or the digital model of care. In presenting the results of the study […].

Also, added to the Discussion, paragraph 5, page 19: “[…] Finally, as noted above, patients self-select into either of the treatment models. This may lead to a risk of selection bias of the cost estimates in that more care needing patients will seek face-to-face treatment. However, we note that the two groups of patients are comparable in terms of sex and age”

Reviewer comment:

Since the patient transportation and time are regarded as the largest costs, the authors should comment on that, e.g. retired patients have no cost for time, and on the choice of vehicle, e.g. electric car. The latter implies lower or no CO2-effect. Have you estimated the cost due to real emission or the compensation cost, which is quite different.

Regarding CO2-emissions, are the servers for the cloud based Joint Academy fossil free?

Would the patient make a concomitant errand when visiting the physiotherapist?

Author response:

In a costing analysis all resources need to be accounted for, including time costs. In line with common practice we also need to value the time of all people, regardless of employment status. In the case of retired persons their time costs should be valued according to the opportunity cost of leisure, which is generally proposed to equal 35% of working time (valued at gross wage, net of employment and social security contributions). As noted above there was no difference in age between the two groups of patients allowing us to apply the same valuation of time costs to patients in both models.

Regarding the type of vehicle, data suggest that electric vehicles are still comparably rare in Sweden and there is no reason to assume that it would be more common among this group of patients. Hence, no adjustments are made for the type of vehicle. CO-2 emission costs are estimated by real estimated emissions, not compensation costs. Possible CO-2 emissions from the computer servers driving the JA model have not been included, but nor have such emissions from, e.g., the heating of facilities used in the traditional model due to lack of data. For the same reason, no adjustments have been made for the possibility of dual-purpose transportations. These kinds of adjustments are rarely made in costing analyses unless there are strong reasons to believe that they may affect the final results.

Author action:

None

Reviewer comment:

Discussion: You write "So far, no trials have been published comparing outcomes from face-to-face programs with digital equivalents." Do you refer the joint problems or other areas? In fact, there are numerous RCT-studies in depression - mostly from Sweden - treated with cognitive based therapy (CBT) on internet.

Author response:

That is correct, we are referring to osteoarthritis here.

Author action:

Added the word “OA” in front of “[…] face-to-face programs […]” in the Discussion, paragraph 3, page 17.

Reviewer comment:

What is the impact of the platform vs the physiotherapist, which might influence the costs? Will self-care suffice for some patients, e.g. even cheaper?

To optimize the cost savings, how can patients be selected in the future?

Author response:

According to all international recommendations, OA should primarily be treated with education on the disease, supervised and individualized exercise, and if needed help with weight control. It is, as we know from the literature and perhaps also from personal experience, hard to change behavior (performing individualized exercises for the purpose of improving strength and mobility, on a regular basis). Especially if the individual receives no professional guidance or support. So in this case, for the majority of cases self-care is not a viable option.

In terms of optimizing the cost savings, it is expected that patients will continue to self-select, but with increasing support and advice from their care providers to make the most appropriate choice between all treatment alternatives. 

Author action:

None

Reviewer comment:

In the declaration for conflict of interest, it should be mentioned that Arthro Therapeutics owns the JointAcademy Platform.

Author response:

Thanks for highlighting this, it will be clarified in the Conflict of Interest text.

Author action:

Added the following text to the Conflict of Interest Statement: “[…] and Chief Medical Officer of Arthro Therapeutics®, which owns the Joint Academy platform […] ”.

Reviewer #2 

Reviewer comment:

Overview: In this study the authors present the results of a costing tool that provides a comprehensive analysis of a new digital platform for delivering first-line care including disease information and physiotherapy to patients with osteoarthritis (OA) and compare this with an existing face-to-face model of treatment. This is an original study and it provides new and valuable information in the area of digital health. The manuscript is very well written and presented. The cost comparison and cost-effectiveness analysis of digital and face-to-face modes of delivery of structured first-line treatment for knee and hip OA strongly supports the idea that digitally delivered care can substantially decrease the economic burden of OA for patients and health care providers. Patients with OA can still benefit from face-to-face interactions with their health care providers but in terms of monitoring and follow-up digital tools are the way forward for the future.

This reviewer does not have any specific comments for major or minor revisions. The major papers that have been published on the global burden of OA have all been cited in the office have done a very good job of leading in this niche but very important area of cost effectiveness research.

Author response:

Thank you for your positive and encouraging comments and for taking the time to review the manuscript.

Author action:

None.

---

## [Editor Report · Decision Letter 1]

7 Jul 2020

Costing analysis of a digital first-line treatment platform for patients with knee and hip osteoarthritis in Sweden

PONE-D-20-13627R1

Dear Dr. Nero,

We’re pleased to inform you that your manuscript has been judged scientifically suitable for publication and will be formally accepted for publication once it meets all outstanding technical requirements.

Kind regards,

Patrick Bergman

Academic Editor

PLOS ONE
---

## [Editor Report · Acceptance letter]

10 Jul 2020

PONE-D-20-13627R1 

Costing analysis of a digital first-line treatment platform for patients with knee and hip osteoarthritis in Sweden 

Dear Dr. Nero:

I'm pleased to inform you that your manuscript has been deemed suitable for publication in PLOS ONE. Congratulations! Your manuscript is now with our production department. 

Kind regards, 

on behalf of

Dr. Patrick Bergman 

Academic Editor

PLOS ONE